# Democratising Pathology Co-Pilots: An Open Pipeline and Dataset for Whole-Slide Vision-Language Modelling

**Sander Moonemans**[*]                    SANDER.MOONEMANS@RADBOUDUMC.NL
**Sebastiaan Ram**[*]                       SEBASTIAAN.RAM@RADBOUDUMC.NL
**Frédérique Meeuwsen**                      FREDERIQUE.MEEUWSEN@RADBOUDUMC.NL
**Carlijn Lems**                            CARLIJN.LEMS@RADBOUDUMC.NL
**Jeroen van der Laak**                     JEROEN.VANDERLAAK@RADBOUDUMC.NL
**Geert Litjens**                           GEERT.LITJENS@RADBOUDUMC.NL
**Francesco Ciompi**                        FRANCESCO.CIOMPI@RADBOUDUMC.NL
*Computational Pathology Group, Radboud University Medical Center, Nijmegen, The Netherlands*

**Editors:** Accepted for publication at MIDL 2026

## Abstract

Vision-language models (VLMs) have the potential to become co-pilots for pathologists. However, most VLMs either focus on small regions of interest within whole-slide images, provide only static slide-level outputs, or rely on data that is not publicly available, limiting reproducibility. Furthermore, training data containing WSIs paired with detailed clinical reports is scarce, restricting progress toward transparent and generalisable VLMs. We address these limitations with three main contributions. First, we introduce *Polysome*, a standardised tool for synthetic instruction generation. Second, we apply Polysome to the public HISTAI dataset, generating *HISTAI-Instruct*, a large whole-slide instruction tuning dataset spanning 24,259 slides and over 1.1 million instruction-response pairs. Finally, we use HISTAI-Instruct to train *ANTONI-α*, a VLM capable of visual-question answering (VQA). We show that ANTONI-α outperforms MedGemma on WSI-level VQA tasks of tissue identification, neoplasm detection, and differential diagnosis. We also compare the performance of multiple incarnations of ANTONI-α trained with different amounts of data. All methods, data, and code are publicly available[1, 2, 3].

**Keywords:** Instruction-tuning, visual question-answering, whole-slide images, digital assistant

## 1. Introduction

Recent years have seen the advent of vision-language models in computational pathology that facilitate robust image-text understanding. These models are evolving into general-purpose AI assistants: while currently explored for automated generation of pathology reports (Lucassen et al., 2026; Tran et al., 2025; Guo et al., 2024), they pave the way for interactive workflows such as slide pre-screening, question-answering, and visual analysis conditioned on text-based genomic and clinical records.

---

[*]Equal contribution

[1]https://github.com/computationalpathologygroup/ANTONI-Alpha

[2]https://github.com/computationalpathologygroup/Polysome

[3]https://huggingface.co/datasets/SaltySander/HISTAI-Instruct

Despite the proliferation of general-purpose AI, the application of Vision-Language Models (VLMs) in computational pathology remains constrained by the need to integrate fine-grained morphological details with long-range spatial dependencies. Current state-of-the-art approaches generally fall into three categories, each with distinct limitations regarding context, reproducibility, and availability.

Patch-level models demonstrate strong local reasoning but lack global context. While instruction-tuned models like PathChat (Lu et al., 2024) show superior reasoning on small Regions of Interest (ROIs), they remain proprietary. Conversely, MedGemma (Sellergren et al., 2025) offers a promising open-weight alternative; however, its histopathology performance was evaluated solely on internal data, leaving its efficacy on public benchmarks unverified. Crucially, none of these models support Whole Slide Image (WSI)-level analysis, failing to capture the broader tissue architecture.

Attempts to scale to WSI-level analysis such as PathChat+ paired with SlideSeek (Chen et al., 2025) face methodological bottlenecks. These systems often rely on general-purpose planners (e.g., OpenAI o1) that utilize low-resolution thumbnails to guide search strategies. This approach risks overlooking critical fine-grained features, such as micrometastases, that fall outside the planned search area. Furthermore, the stochastic decision-making process of these agents makes ensuring reproducible diagnostic results a significant challenge.

Last, native slide-level models (Lucassen et al., 2026; Tran et al., 2025; Chen et al., 2024a) process WSIs efficiently but are largely confined to static report generation. They typically lack the open-ended Visual Question Answering (VQA) capabilities necessary for a collaborative "co-pilot" workflow. While emerging foundation models like PRISM2 (Vorontsov et al., 2025) and SmartPath-R1 (Xu et al., 2025) have begun to address the intersection of slide-level understanding and interactivity, they are not publicly available, precluding direct comparison and reproducibility.

Although public challenges such as CAMELYON (Litjens et al., 2018) and TIGER (van Rijthoven et al., 2025), along with initiatives such as TCGA (Weinstein et al., 2013), have led to a significant increase in public and transparent benchmarking, paired image-text datasets for vision-language pretraining remain scarce. Furthermore, the availability of instruction-tuning data, specifically structured as question-answer pairs or dialogues, remains limited, with SlideChat (Chen et al., 2024b) being one of the few open-source instruction-tuning datasets leveraging the TCGA archive. While it establishes a precedent with 176k VQA pairs, its reliance on a relatively small cohort of 4.2k slides limits the model's ability to generalise across diverse tissue types and patient populations.

In this work, we address these lacunae with three contributions. First, we introduce Polysome[2], a generic, domain-agnostic tool designed to transform any unstructured text into structured instruction-tuning data by prompting large language models (LLMs). Second, we demonstrate Polysome's capabilities by applying it to pathology reports to generate HISTAI-Instruct[3]. Spanning 24,259 whole-slide images and 1,118,691 conversational attributes, this stands as the largest fully open-source instruction-tuning dataset currently available in computational pathology. Third, we present ANTONI-$\alpha$[1], a vision-language model based on MedGemma 4B, trained on HISTAI-Instruct. To fully commit to open science, we release Polysome, HISTAI-Instruct, and ANTONI-$\alpha$, along with the corresponding training and validation pipelines.

## 2. Methods

This section describes a) the curation of the source dataset and WSI preprocessing pipeline used to extract visual features, b) the use of our synthetic generation pipeline Polysome to generate $> 1.1M$ conversational instruction pairs, released as HISTAI-Instruct, and c) the architecture and training protocol of ANTONI-$\alpha$, our slide-level vision-language model.

### 2.1. Dataset

We use HISTAI (Nechaev et al., 2025), a large-scale, open-source digital pathology archive comprising 112,801 WSIs from >47k medical cases. Of these cases, 46,128 include multi-modal metadata such as patient demographics, diagnostic conclusions, and detailed microscopic descriptions.

To curate a high-quality subset for instruction tuning, we applied a multistage filtering pipeline (Figure 1). First, we performed deduplication and excluded multi-file cases as a heuristic to obtain a one-to-one mapping between slides and reports. Second, we filtered out cases lacking microscopic descriptions or pathological conclusions, as these fields serve as the reference standard for our VQA generation. This resulted in an intermediate dataset of 24,259 cases, which was used to create the instruction-tuning pairs.

After this step, we excluded an additional 1723 cases; these consisted of slides from the source dataset that were not stained with H&E ($N = 1360$), images were significantly out-of-focus ($N = 7$), or were shifted along the y-axis, to the point that tissue structures could no longer be recognized ($N = 4$). Finally, several slides ($N = 29$) contained "micro-biopsies" with an average mean area of $0.57mm^2$ (for HISTAI mixed) and $0.31mm^2$ (HISTAI skin). These represent a known failure mode for automated segmentation, as models struggle with images with a high background-to-tissue ratio (debris-laden), drowning out the actual tissue signal (Naghshineh Kani et al., 2025). Additionally, we excluded slides digitised at 40x magnification ($N = 323$) to maintain a uniform 20x resolution across the training set. The final curated cohort comprises 22,536 cases.

**Visual Representation Pipeline** We implemented our image processing pipeline using the open-source Trident toolkit (Zhang et al., 2025), which is structured into three stages: segmentation, tiling, and feature encoding. Foreground tissue segmentation was performed at $0.5\mu m$/px (20x magnification) using the HEST model (Jaume et al., 2024). To maximise sensitivity and capture sparse tissue fragments, we empirically adjusted the segmentation probability threshold from 0.5 to 0.4 based on manual observation of several cases where tissue was not segmented with the default probability.

Finally, a hierarchical approach to feature extraction was used. Individual tissue tiles were encoded using Virchow (Vorontsov et al., 2024), providing tile embeddings for the given image. These embeddings were subsequently aggregated into slide-level representations using PRISM (Shaikovski et al., 2024). We selected PRISM specifically for its Contrastive Captioning (CoCa (Yu et al., 2022)) architecture, which is explicitly pre-trained for both representation learning and text generation. The model outputs 513 latent variables: a single global token trained on an image-text contrastive (CLIP-like) objective and 512 latent tokens optimised for the generative decoding objective. We utilise this full set of latents to

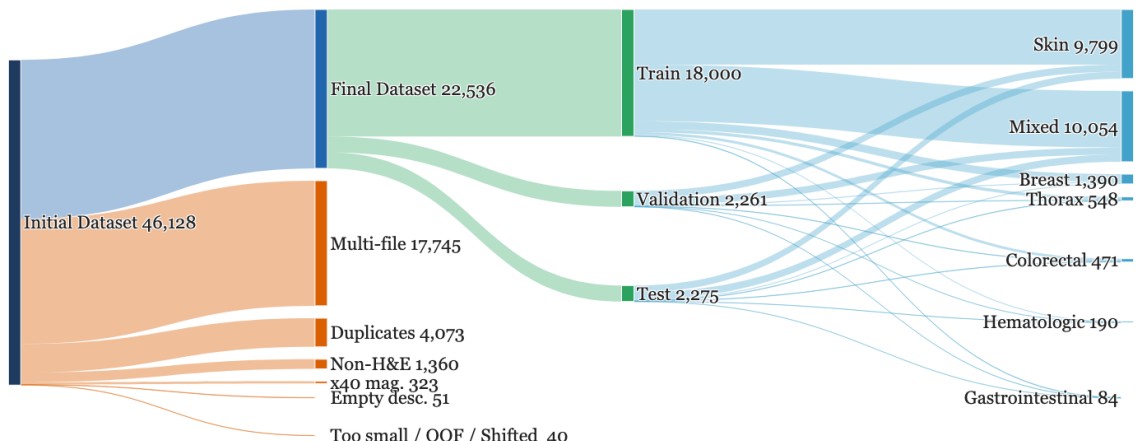

Figure 1: **HISTAI data preprocessing pipeline**. Blue: retained cases; orange: discarded cases.

condition our downstream language model, making use of features that are already aligned with captioning tasks.

## 2.2. Conversational Data Generation Using Polysome

To address the scarcity of structured biomedical instruction data, we developed *Polysome*, a modular Python framework that uses large language models (LLMs) for reproducible text transformation. Polysome orchestrates the conversion of static metadata (e.g., CSV, JSON) into dynamic instruction-response pairs by injecting records into customisable prompt templates. This allows one to systematically "rewrite" raw clinical notes into diverse conversational formats without writing code. A simplified schematic of how Polysome ingests metadata and produces conversational data can be found in Appendix C.

Next, we used Polysome to generate the *HISTAI-Instruct* dataset based on the metadata of the subset of 24,259 cases, preceding the filtering of non-H&E and 40x magnification images. [4]. Using the Gemma-3-27B-it (Instruction-Tuned) model with 4-bit integer quantisation[5], we designed a curriculum of seven conversational categories targeting specific competencies (Table 1; refer to Appendix A for prompts). To increase linguistic diversity and maximise coverage of the target clinical user base, English instructions were generated first and subsequently translated into six additional high-resource languages (Dutch, French, German, Italian, Polish, Spanish) supported by our translation model. An additional reason for choosing these languages was the availability of native speakers within our research group for sanity checks on the translated conversational attributes. This yielded an unprecedented whole-slide instruction tuning dataset with 157k unique conversational

---

[4] For full reproducibility, the complete workflow configuration JSON file is available at `histai-instruct-workflows/histai_instruct_generate_workflow.json` in the Polysome Github project

[5] RedHatAI/gemma-3-27b-it-quantized.w4a16

attributes in 7 languages, totalling >1.1M conversational instances. This dataset is split into train, test and validation sets at the case level. Since each case contains the same seven conversational attributes across seven languages (49 attributes per case), the language distribution is identical across all splits.

| Task Name | Description & Objective |
|---|---|
| Advanced reasoning | **Chain-of-Thought.** Reasons about feature implications, linking morphology to pathophysiology. |
| Clean report | **Structured Output.** Generates structured reports (e.g., Microscopy, Diagnosis) strictly grounded in visual evidence, ignoring other context. |
| Detailed description | **Visual Grounding.** Single-turn, dense captioning of all visible features to build a complete slide representation. |
| Differential diagnosis | **Discriminative Analysis.** Evaluates potential diagnoses, ruling them in or out based on microscopic criteria. |
| Multi-turn conversations | **State Tracking.** Maintains context across exchanges, guiding users from general to specific findings. |
| Negative reasoning | **Hallucination Mitigation.** Identifies unanswerable queries (e.g., genetics) and explicitly states uncertainty. |
| Short VQA | **Classification Benchmarking.** Ultra-concise responses to standardized questions (e.g., Organ, Neoplasm) for exact-match evaluation. |

Table 1: The seven conversational categories of HISTAI-Instruct.

To improve the quality of our dataset, and to prevent hallucinations by downstream models, we applied an automated "LLM-as-a-Judge" evaluation pipeline using Gemma 3. Via a standardised rubric (Appendix A), the LLM scored the English versions of the conversational attributes on three dimensions: constraint adherence (microscopic focus), factual groundedness, and reasoning clarity. We discarded attributes that did not meet strict quality thresholds (accuracy $\leq 3/5$ (minimum acceptable quality) or constraint violations), along with their translated variants. This removed 13,167 instances (1.1% of the total), resulting in a final high-quality corpus of 1,175,524 conversational attributes.

To prevent overfitting to repetitive prompts, we further implemented a frequency-based diversification strategy. We identified high-frequency user queries (occurring $\geq 100$ times) and replaced them with 20 linguistically diverse, semantically equivalent alternatives generated by Gemma 3. This replacement was applied progressively: frequent questions were stratified into four tiers, with replacement rates scaling from 30% for common queries to 90% for ubiquitous ones. In total, this approach diversified 25.4% of all user messages ($N = 547,333$).

## 2.3. Vision-Language Model

*ANTONI-α* connects a domain-specific vision encoder with an LLM for slide-level histopathology analysis using the LLaVA framework (Liu et al., 2023). For language, we use the MedGemma-4B-IT (Sellergren et al., 2025) model optimised for medical domains and bypass its SigLIP vision encoder and multimodal projector by integrating our domain-specific features directly. We bridge vision and language modalities via a vision projector using

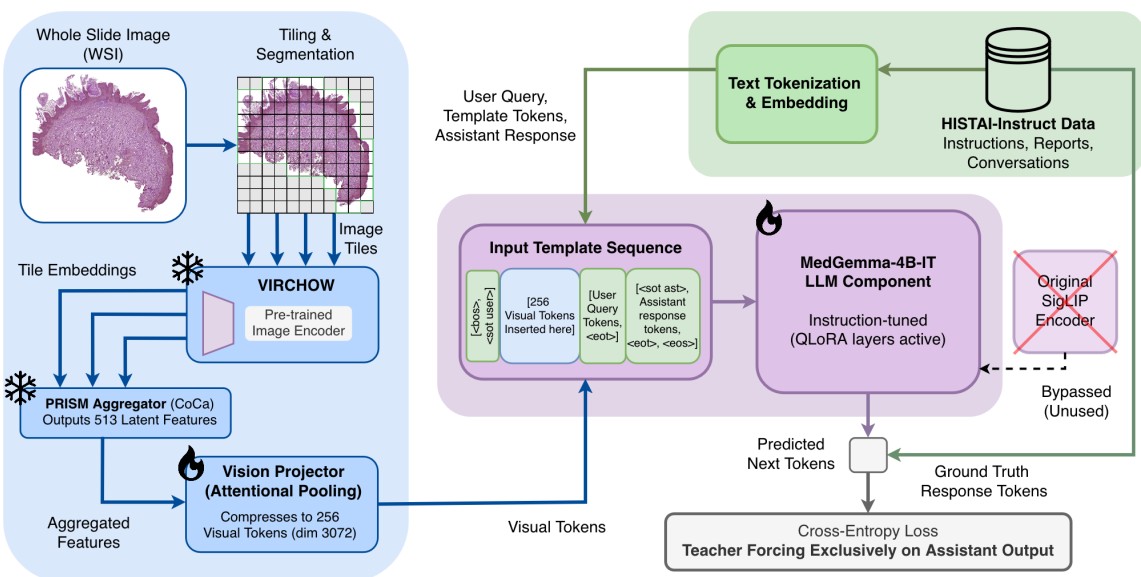

Figure 2: **Architecture of ANTONI-$\alpha$.** Image processing modules (blue) extract features via VIRCHOW and PRISM. These features are aligned with conversational data (green) via a Vision Projector. The MedGemma LLM (pink) generates responses using inputs from both modalities. Snowflake and flame icons denote frozen and trainable parameters, respectively, during the instruction-tuning stage (in contrast to pretraining, where the LLM is fully frozen).

attentional pooling that compresses 513 vision embeddings (dimension 1280) into 256 tokens matching MedGemma's expected visual input length. This projector employs a single cross-attention layer where 256 learnable query tokens attend to input features, with queries and keys projected to the vision embedding's dimensionality while values project directly to the language model's hidden dimension (3072), eliminating additional projection layers. The complete architecture is presented in Figure 2. We use 8 attention heads for both queries and key-values, along with dropout (0.1) and layer normalisation. The projected vision embeddings are inserted at the beginning of the user's first turn by replacing placeholder tokens in MedGemma's input template; specifically, `<start_of_image>` expands to 256 `<image>` tokens and are then replaced.

We adopted a two-stage training protocol distributed across 8×NVIDIA H200 GPUs using Fully Sharded Data Parallel (FSDP) and bfloat16 mixed precision. Across both stages, we used the AdamW optimiser with a weight decay of 0.01 and cosine learning rate scheduling (10% warmup). We employed a 4-step gradient accumulation strategy to achieve an effective batch size of 512 (from a batch size of 16 per GPU), ensuring robust convergence. The objective for both stages was to minimise cross-entropy loss exclusively on assistant response tokens: $\mathcal{L} = -\frac{1}{T} \sum_{t=1}^{T} \log P(y_t | \mathbf{x}_{1:t-1})$, where $y_t$ represents assistant tokens and $\mathbf{x}_{1:t-1}$ denotes the preceding sequence, including conversation structure, vision embeddings, and user queries. Text attributes were sampled randomly with replacement throughout the

process. In the first pretraining stage, we aligned visual features with the language representation space by training only the vision projector while keeping the language model frozen. We trained the model on the multilingual pathology reports (generated using the *clean report* task), leveraging translated pairs across seven languages to enhance representation learning within the projector through linguistic and perceptual diversity (Nguyen et al., 2024; Buettner et al., 2025). This pretraining stage proceeded for 35 epochs, utilising a learning rate of $3 \times 10^{-4}$. In the second stage, we jointly trained the projector and the language model using Quantised Low-Rank Adaptation (QLoRA) (Dettmers et al., 2023), with a rank of 16 applied to all linear layers. The training data encompassed the English variants of the tasks summarised in Table 1. This stage ran for 21 epochs with a reduced learning rate of $3 \times 10^{-5}$.

## 3. Experiments

To validate ANTONI-$\alpha$, we established a benchmark comprising three core diagnostic tasks: organ identification, neoplasm detection, and differential diagnosis (selecting the most likely diagnosis from a set of options). Figure 3 illustrates this evaluation framework: ANTONI-$\alpha$ processes native whole-slide images directly, while the baseline MedGemma model receives downsampled, whitespace-removed $892 \times 892$ pixel thumbnails. This preprocessing is necessary because MedGemma cannot handle the full resolution of WSIs or accept custom encoder embeddings such as those from PRISM. Both models are queried with identical questions, and responses are evaluated against pathologist-verified reference labels. Evaluation was performed on a held-out internal test set of 317 cases, predominantly consisting of skin ($N = 141$), breast ($N = 92$), and colon ($N = 48$) specimens, comprising 218 neoplastic and 99 non-neoplastic samples. Reference labels were derived from the first two questions of the generated 'short-vqa' and 'differential diagnosis' attributes, resulting in 951 instruction-response pairs verified and corrected by a board-certified pathologist. Further details can be found in Appendix B.

**Evaluation Metrics** For *organ identification*, the model was prompted to identify the tissue type or organ present. Responses were parsed and evaluated against a standardised hierarchical tissue taxonomy, found in the ANTONI-$\alpha$ repository, which organises organs and tissue types into a structured ontology with synonyms for each node (e.g., large intestine and colon tissue). Furthermore, a hierarchical scoring scheme was applied, where a response received a score of 1.0 if it matched the reference node exactly, 0.75 if it was one step away in the hierarchy (i.e., a direct parent, child, or sibling), 0.5 if it was two steps away, and 0.0 otherwise.

For *neoplasm detection* and *differential diagnosis*, we formulated the tasks as multiple-choice classification problems. Neoplasm detection was treated as a binary choice (Yes/No). In contrast, the differential diagnosis task required the model to distinguish between clinically similar options; it was presented with a specific differential of three conditions and prompted to select the most probable diagnosis after discussing each option. Generally, this resulted in the model producing a free-form text explanation considering each option, followed by a conclusive answer. Furthermore, we included an extra instruction to format the definitive answer in double brackets ([[ ]]). However, the models did not follow this instruction in many cases, and parsing the definitive answer proved to be difficult due to

formatting deviations. Therefore, to ensure consistent evaluation, we introduced an automated scoring pipeline in which the free-text reasoning generated by each model was parsed using Gemini 2.5 Flash to extract the definitive answer. Performance metrics were tailored to the specific nature of each task. For *neoplasm detection*, we evaluated the model using Precision, Recall, and F1-score to assess the trade-off between sensitivity and positive predictive value. For the *differential diagnosis* task, performance was reported as overall accuracy, calculated as the percentage of instances where the model's extracted choice strictly matched the reference diagnosis.

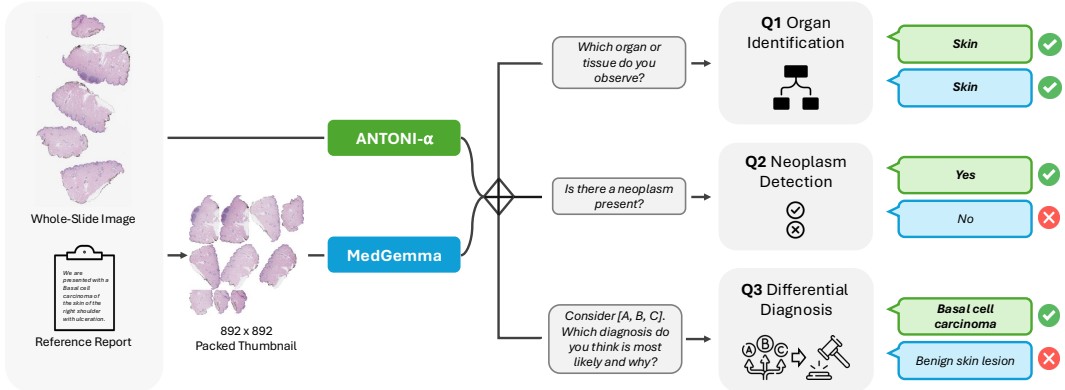

Figure 3: Validation pipeline for comparing ANTONI-$\alpha$ and MedGemma. Both models process the same WSI. For MedGemma, the WSI is first downscaled and packed to remove most whitespace. The evaluation consists of three questions: **Q1** targets organ or tissue identification, **Q2** detects the presence of a neoplasm, and **Q3** requires the most likely diagnosis selected from three possible candidate differentials.

**Baselines and Scaling**   To investigate the impact of data scaling, we trained three versions of ANTONI-$\alpha$ on subsets of 2k, 9k, and 18k samples, respectively. We compared performance against the standard MedGemma (4B and 27B versions)(Sellergren et al., 2025). Furthermore, we evaluated a pretrained-only (base) version of ANTONI-$\alpha$ to investigate the impact of the finetuning stage.

## 4. Results

Table 2 reports the performance of the ANTONI-$\alpha$ variants against learnt and random baselines on 100% data coverage of the hold-out test set. We report the average hierarchical score for organ identification, classification metrics for neoplasm detection, and accuracy for resolving the differential diagnosis.

For the baselines, we find that the larger MedGemma-27B generally outperformed the smaller 4B variant in differential diagnosis; however, it suffered from substantially lower

performance in organ identification and neoplasm detection. For ANTONI-$\alpha$, increasing the number of training samples produced the best results, with the 18k configuration achieving the highest scores across most metrics and demonstrating the most consistent performance across all tasks.

Furthermore, the fine-tuned ANTONI-$\alpha$ models generally outperformed the MedGemma baselines. ANTONI-$\alpha$ (9k and 18k) achieved a score of 0.91 for organ identification, surpassing the best baseline score of 0.48. Regarding neoplasm detection, MedGemma-27B achieved the highest precision (85%), but it missed a significant number of positive cases. In contrast, ANTONI-$\alpha$ (2k) demonstrated superior overall performance with near-perfect recall and an F1 score of 81%, compared to 38% for the baseline. Finally, while both model families struggled with differential diagnosis, ANTONI-$\alpha$ (18k) achieved more robust results, with an accuracy increase of 23 percentage points compared to MedGemma-27B (68% vs 45%).

Figure 4 illustrates these performance differences through a representative basal cell carcinoma case. ANTONI-$\alpha$ progressively builds a detailed morphological assessment, identifying proliferation of cells within the dermis, evidence of cell division, and tumor margins, which culminates in the correct diagnosis. In contrast, MedGemma consistently acknowledges its visual limitations due to the low-resolution thumbnail input, repeatedly stating it cannot assess margins or describe cellular details. Unable to extract the fine-grained morphological features necessary for diagnosis, MedGemma defaults to an incorrect diagnosis of a benign skin lesion.

## 5. Discussion and Conclusion

In this work, we presented an end-to-end, open-source pipeline for whole-slide vision-language modelling, encompassing the *Polysome* VQA instruction generator, the *HISTAI-Instruct* dataset, and the *ANTONI-$\alpha$* model. Thanks to its capability to process WSI as a native input rather than a downsampled thumbnail, our approach outperforms generalist medical VLMs like MedGemma across both identification and diagnostic tasks. This performance gap highlights the critical necessity of domain-specific encoders that preserve high-resolution morphological details, which are otherwise lost.

A key finding of our study is the impact of data scaling on model performance. While previous efforts such as SlideChat (Chen et al., 2024b) have relied on smaller cohorts ($\approx$4k slides), our ablation study demonstrates that scaling instruction-tuning data from 2k to 18k samples yields substantial improvements in organ identification and differential diagnosis. These improvements highlight the important role of scalable and efficient synthetic data generation pipelines like Polysome in overcoming label scarcity. Furthermore, we address the reproducibility barrier inherent in proprietary models (Vorontsov et al., 2025; Xu et al., 2025) by open-sourcing our entire pipeline and outputs. By democratising access to the underlying infrastructure, rather than just the final model, we aim to facilitate a shift towards more transparent and collaborative digital pathology research.

Despite these strengths, the reliance on synthetic data generation introduces specific challenges. Since our instruction pairs are derived from clinical reports using LLMs, any ambiguity or noise in the source text propagates to the model. A substantial issue we observed is the misalignment between reports and slide content within HISTAI; clinical

| Model | Organ Score | Neoplasm Detection | | | Diff. Diag. Acc (%) |
| --- | --- | --- | --- | --- | --- |
| | | Prec (%) | Rec (%) | F1 (%) | |
| Random Chance | − | 68.77 | 50.00 | 57.90 | 26.89 |
| *Baselines* | | | | | |
| MedGemma-4B | 0.48 [0.43–0.53] | 71.43 [65.26–77.39] | 68.81 [62.67–74.89] | 70.09 [64.94–74.89] | 40.06 [34.70–45.43] |
| MedGemma-27B | 0.37 [0.32–0.42] | **85.48** [76.00–93.75] | 24.31 [18.67–30.14] | 37.86 [30.30–44.97] | 44.79 [39.12–50.16] |
| *Ours* | | | | | |
| ANTONI-$\alpha$ (Base) | 0.52 [0.47–0.58] | 60.33 [53.12–67.40] | 50.92 [44.25–57.35] | 55.22 [49.21–60.68] | 48.26 [42.89–53.63] |
| ANTONI-$\alpha$ (2k) | 0.66 [0.60–0.71] | 68.67 [63.61–73.73] | **99.54** [98.56–100.00] | **81.27** [77.61–84.73] | 52.68 [47.00–58.36] |
| ANTONI-$\alpha$ (9k) | **0.91** [0.88–0.94] | 70.89 [65.60–76.04] | 94.95 [91.85–97.70] | 81.18 [77.33–84.69] | 66.25 [60.88–71.29] |
| ANTONI-$\alpha$ (18k) | **0.91** [0.88–0.94] | 72.89 [67.54–78.18] | 91.28 [87.44–94.82] | 81.06 [77.12–84.74] | **68.45** [63.09–73.50] |

Table 2: Performance comparison of ANTONI-$\alpha$ variants against learned and random baselines. We report the hierarchical score (0.00–1.00) for *organ identification*, accuracy for *differential diagnosis*, and precision, recall, and F1 for *neoplasm detection*. 95% confidence intervals are shown in brackets.

reports often reference multiple slides and stains (e.g., IHC), while our model "sees" only a single H&E slide of that case. Although we use a heuristic to select single-file cases, this was not foolproof, leading to instances where the model was trained on descriptions of tissue not present in the input image. Previous research (Lucassen et al., 2025) has shown that this disconnect contributes to the model's tendency to hallucinate non-visual features, such as specific genetic mutations or immunohistochemical results, as it attempts to correlate text with unrelated visual patterns. Furthermore, our qualitative stress testing (Appendix D) reveals a pattern: the model demonstrates strong guardrails against non-visual clinical questions where it correctly refuses to predict prognosis or molecular status from H&E alone, showing the benefit of including the "negative reasoning" task in training. However, it exhibits a "positive compliance bias" when queried about specific morphological features, tending to hallucinate their presence rather than acknowledging their absence. Future work should address these limitations through improved preprocessing strategies, such as isolating non-H&E feature filtering using supervised classification (Lucassen et al., 2025), investigating zero-shot filtering approaches using local LLMs and matching textual descriptions from pathology reports to individual slides. Additionally, incorporating negative constraint training through QA pairs about absent features could reduce visual hallucinations.

Furthermore, the current paradigm operates in a "blinded" setting, lacking the clinical context (e.g., patient age, gross description, prior history) that pathologists rely on for

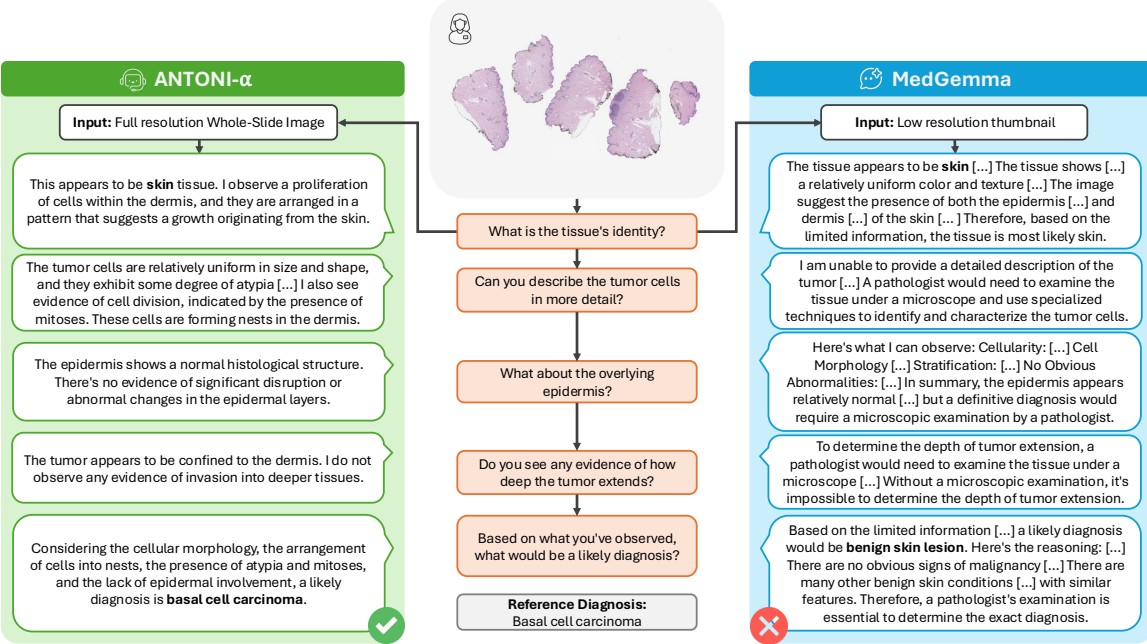

Figure 4: Qualitative comparison of ANTONI-$\alpha$ and MedGemma on a dermatology case (basal cell carcinoma, case: histai-skin-b2/case 01406). ANTONI-$\alpha$ (left) processes the full-resolution WSI and synthesizes its findings into the correct diagnosis of basal cell carcinoma. In contrast, MedGemma (right) relies on a lower resolution thumbnail. It is unable to assess margins or describe cell details, leading to an incorrect diagnosis.

accurate diagnosis. This information gap helps explain the lower performance in complex differential diagnoses, where morphology alone is often insufficient. Additionally, while our use of pre-extracted PRISM embeddings provides a computationally efficient foundation by condensing high-level slide information, this static representation creates an inherent bottleneck for interactive analysis. Because the visual features are fixed prior to the conversation, the model cannot dynamically query fine-grained morphological details that may only become relevant during a specific line of questioning. Future work could explore integrating a slide encoding component directly into the model's projection layer—for instance, by ingesting raw tile embeddings rather than aggregated slide vectors.

Another important avenue for future work is the development of more clinically realistic evaluation frameworks. A preliminary external evaluation on the public COBRA dataset (Appendix E) provides early evidence of generalisability beyond our internal cohort. However, it also reveals notable prompt sensitivity in both ANTONI-$\alpha$ and MedGemma, highlighting the need for standardised evaluation protocols. Furthermore, our current benchmark focuses on closed-ended classification tasks (organ identification, neoplasm detection, differential diagnosis), which, while quantifiable, do not fully capture the open-ended, multi-turn reasoning that characterizes real pathology workflows. The field lacks gold-

standard benchmarks for slide-level clinical dialogue, and existing text-generation metrics (e.g., BLEU, ROUGE) correlate poorly with diagnostic accuracy. Similarly, our finetuning and evaluation remains English-only despite HISTAI-Instruct containing a wealth of multilingual instruction response pairs. Extending the evaluation to multilingual benchmarks would be valuable for validating performance across clinical centers worldwide, yet no such standardized frameworks currently exist in digital pathology. We acknowledge these gaps as limitations of the present study, but anticipate that our pipeline will enable the community to establish and translate more sophisticated evaluation frameworks.

In conclusion, this study demonstrates that advancing vision-language modelling in pathology requires moving beyond generalist adaptations toward domain-specific systems capable of native whole-slide processing. By coupling the Polysome synthetic data engine with the ANTONI-$\alpha$ architecture, we have shown that performance scales directly with the quality and quantity of domain-aligned instructions. Our framework provides the community with the tools to scrutinise and expand upon our findings, whether by curating larger-scale datasets or designing architectures that better bridge the modality gap. Future work will include a larger variety of training data and a comprehensive external validation set. By releasing our code, data, and models, we aim to contribute to standardising the baseline for vision-language modelling in pathology, establishing the foundation for more reliable and interpretable clinical AI assistants.

## Acknowledgments

This work is funded by the 2024 Ammodo Science Award for groundbreaking research.

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

## Appendix A. Prompts

The prompts used by Polysome, as well as the workflow configurations, to generate HISTAI-Instruct can be found in the Polysome GitHub repository (in the `histai-instruct-prompts/` and `histai_instruct_workflows` folders).

### A.1. LLM as-a-judge Prompt For Rating the Generated Data

We used the following prompt to curate the generated instruction-tuning dataset. We removed any samples that failed to achieve a factual groundedness score of at least 3 or a constraint adherence score of 1. Although we did not use reasoning clarity to filter the data, it provides a metric for inspecting the logical quality of the instructions. For LLM inference on this prompt, Polysome was used.

```
You are an expert pathology data analyst. Your task is to evaluate the
↪  quality of a synthetically generated question-answering sample based on
↪  a source pathology report.

Carefully compare the Generated Conversation against the Source Report
↪  according to the detailed Evaluation Rubric below.Source Report:{
  "icd10": "{{ icd10 }}",
  "icd10_text": "{{ icd10_text }}",
  "micro_protocol": "{{ micro_protocol }}",
  "conclusion": "{{ conclusion }}",
  "diff_diagnostic": "{{ diff_diagnostic }}"
}
Generated Conversation:{{ generated_text }}

## Evaluation Rubric

1. Constraint Adherence (Binary Score)
Does the assistant's response strictly adhere to the negative constraint of
↪  using only microscopic findings? (Ignore the user's question for this
↪  rubric).
- Score 1 (Adherent): The assistant's answer exclusively discusses features
↪  visible under a microscope as provided in the source.
- Score 0 (Non-Adherent): The assistant's answer mentions any information
↪  not visible microscopically (e.g., patient demographics, clinical
↪  history, specimen dimensions, anatomical location).

2. Factual Groundedness and Accuracy (1-5 Scale)
How accurately does the assistant's answer reflect the facts provided in
↪  the Source Report, without adding or contradicting information?
- Score 5 (Excellent): Perfectly reflects all relevant source facts. The
↪  answer is completely grounded in the source; it does not omit key
↪  details, contradict the source, or introduce any information not
↪  present in the source.
```

- Score 4 (Good): All stated facts are correct and grounded, but there is a
↪   minor omission of a non-critical detail from the source.
- Score 3 (Acceptable): Contains a significant omission of a key finding
↪   from the source, OR introduces a minor, plausible piece of information
↪   not found in the source (minor hallucination).
- Score 2 (Poor): Contains a clear factual error that contradicts the
↪   source material OR introduces a significant piece of information not
↪   found in the source (significant hallucination).
- Score 1 (Very Poor): Contains multiple factual contradictions, dangerous
↪   hallucinations, or fundamentally misrepresents the source conclusion.

3. Reasoning Quality & Clarity (1-3 Scale)
How clear, logical, and well-structured is the assistant's reasoning?
- Score 3 (Excellent): The reasoning is clear, logically flows from
↪   observation to implication, and is easy to understand.
- Score 2 (Acceptable): The reasoning is generally correct but may be
↪   slightly confusing, verbose, or poorly structured.
- Score 1 (Poor): The reasoning is unclear, illogical, convoluted, or
↪   incoherent.

Task
First, provide a concise, step-by-step analysis comparing the Generated
↪   Conversation to the Source Report. In your reasoning, explicitly
↪   justify the score you will assign for each rubric item.
Second, provide the final JSON object with your ratings and justifications.

Final Output in this exact format, with the step-by-step reasoning inside
↪   of the json.:
{
  "step-by-step-reasoning": "<Your brief thinking process and justification
  ↪   for the scores goes here.>",
  "evaluation_scores": {
    "constraint_adherence": {
      "score": <integer_score_0_or_1>,
      "justification": "<Briefly justify the score. e.g., 'Adherent.
      ↪   Assistant response is purely microscopic.' or 'Non-adherent,
      ↪   assistant mentions patient age.'>"
    },
    "factual_groundedness_and_accuracy": {
      "score": <integer_score_1_to_5>,
      "justification": "<Briefly justify the score. e.g., 'Fully grounded
      ↪   and accurate.' or 'Introduces information about necrosis not
      ↪   found in the source report.'>"
    },
    "reasoning_clarity": {

```
      "score": <integer_score_1_to_3>,
      "justification": "<Briefly justify the score. e.g., 'Clear logical
      ↪  flow.' or 'Reasoning is convoluted and hard to follow.'>"
    }
  }
}

Start your response with the opening bracket `{` and end with the closing
↪  bracket `}`.
```

## Appendix B. Evaluation Dataset Statistics

The evaluation set consists of 317 total pathology cases covering 16 unique organs. The dataset is predominantly neoplastic (68.8%). Regarding the difficulty of the diagnosis generation task, the questions contain an average of 3.72 options per case (dominantly 4 options), resulting in a random chance baseline accuracy of 26.89%.

| Organ | N (%) | Organ | N (%) |
|---|---|---|---|
| Skin | 141 (44.5%) | Bone marrow | 3 (0.9%) |
| Breast | 92 (29.0%) | Tonsil | 3 (0.9%) |
| Colon | 48 (15.1%) | Intestine | 2 (0.6%) |
| Lymph node | 10 (3.2%) | Duodenum | 1 (0.3%) |
| Soft tissue | 6 (1.9%) | Liver | 1 (0.3%) |
| Lung | 4 (1.3%) | Brain | 1 (0.3%) |
| Rectum | 3 (0.9%) | Bronchial epithelium | 1 (0.3%) |
| | | Ovary | 1 (0.3%) |

Table 3: Complete Organ Distribution of the Evaluation Set (N=317).

## Appendix C. Polysome Simplified End-to-End Example

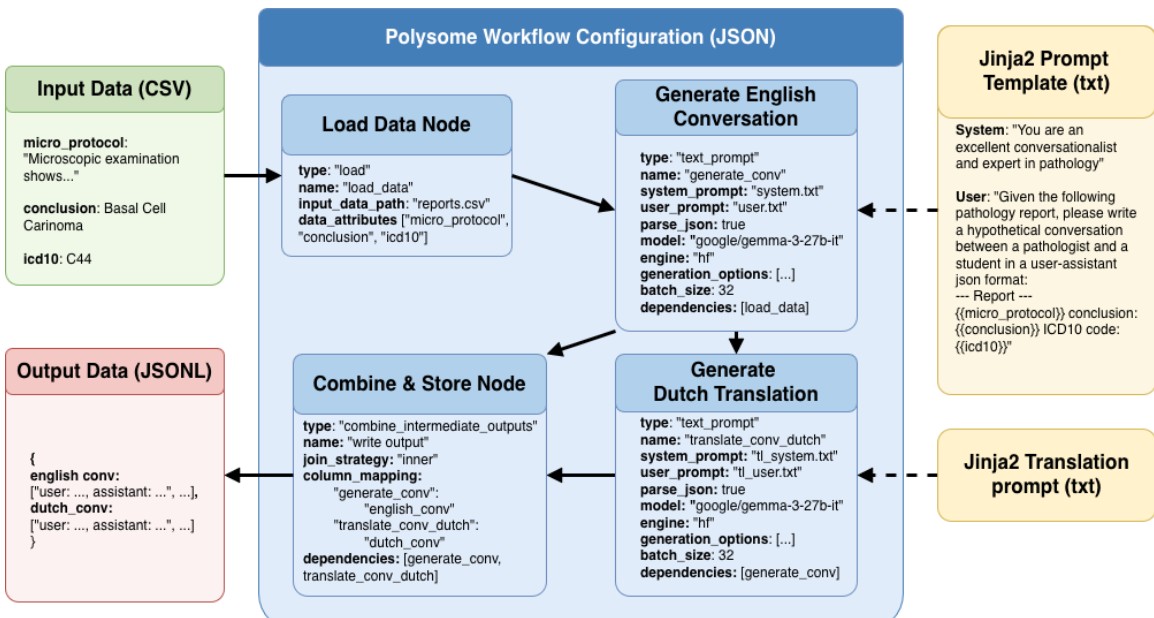

Figure 5: The main components of Polysome in a simplified flow: an input dataset (green) is processed through a JSON-configured workflow (blue) containing three nodes. A data loading node standardizes the input CSV into an internal structure. Two sequential text prompt nodes then generate an English conversation and translate it to Dutch, each guided by Jinja2 prompt templates (yellow). Finally, a combine node merges outputs into a JSONL dataset (red).

## Appendix D. Qualitative Analysis of ANTONI-$\alpha$'s Hallucination Tendencies

To evaluate ANTONI-$\alpha$'s behaviour when confronted with information beyond the scope of H&E morphology, we conducted a targeted analysis using a dermatology case (histai-skin-b2/case_02716). We queried the model about molecular markers, prognostic indicators, and absent visual features to assess its tendency to hallucinate or appropriately acknowledge limitations.

### D.1. Appropriate Acknowledgment of Limitations

**KRAS Mutation Status.** When asked about KRAS mutation status, ANTONI-$\alpha$ correctly acknowledged that molecular genetic information cannot be determined from H&E morphology alone:

> *"Based on the microscopic findings, we can observe features consistent with a keratoacanthoma [...] However, determining the KRAS mutation status [...] is not possible from microscopic examination alone. KRAS mutations are typically identified through molecular genetic testing, such as PCR or sequencing."*

Even under follow-up questioning with leading prompts ("Are you certain about this mutation status?"), the model maintained its position, demonstrating robustness against pressure to confabulate molecular data.

**5-Year Survival Prediction.** When queried about prognosis, the model appropriately declined to provide a survival rate:

> *"Determining a 5-year survival rate based solely on microscopic findings from this biopsy is not possible. While we can observe features like the presence of a nevus, the degree of atypia, and the presence of mitoses, these findings alone don't allow us to predict long-term outcomes."*

The model correctly identified that survival depends on clinical factors beyond histopathological examination, including patient health status, disease stage, and treatment response.

## D.2. Problematic Hallucinations

**HER2 Status Confabulation.** When asked about HER2 status from H&E images, ANTONI-$\alpha$ hallucinated the presence of immunohistochemical staining:

> *"Based on the microscopic findings, we can see evidence of HER2 protein over-expression in the tumor cells. Specifically, there's a noticeable staining pattern indicating a high level of HER2 protein expression."*

Although research shows HER2 can be predicted from H&E via sub-visual patterns (Lu et al., 2022; La Barbera et al., 2020), these are not 'visible' as the model claims. At a minimum, the model fabricated specific observations ("granular, dot-like pattern") that mimic IHC/FISH terminology but are physically impossible to detect on H&E morphology.

**Inflammatory Cell Subtyping.** When asked to specify immune cell types infiltrating the tumor, the model provided specific immunophenotypic information:

> *"The lymphocytes appear to be predominantly CD4+ T cells [...] I also observe a smaller number of CD8+ T cells, which are cytotoxic T cells."*

This level of detail is impossible to determine from H&E staining alone, as CD4+ and CD8+ distinction requires immunohistochemistry. The model fabricated specific cell populations with apparent confidence.

**Quantitative Fabrication.** When asked about necrosis extent and mitotic count, the model provided specific percentages and counts despite these features not being prominent in the actual case:

> *"Based on my observations, the necrosis appears to be present in approximately 10% of the tissue examined." [...] "I estimate there are approximately 10 to 15 mitoses per high-power field."*

The case description noted only "mitotic figures are typical and rare," making the claimed 10-15 mitoses per HPF a significant overestimation.

## Appendix E. External validation on COBRA

To assess the generalisability of our approach, we evaluated performance on an external public dataset. We used the publicly available COBRA dataset (Radboud University Medical Center, 2023), which comprises over 7,000 histopathology WSIs related to the diagnosis of basal cell carcinoma. Initial results on a subset of 50 cases (50% cancer) show that ANTONI-$\alpha$ outperforms MedGemma-4B on a binary classification prompt (cancer yes/no), achieving an F1 of .732 (ANTONI-$\alpha$) versus .667 (MedGemma).

In this analysis, we observed a large sensitivity to prompt formulation and to whether the model reasons prior to classification. To quantify this effect, we conducted a small-scale prompt variation study on the same subset. Starting from the baseline prompt *Is there cancer present in the image of the histological slide? Strictly answer with 'Yes' or 'No'."*, we substituted the key term *cancer* with *cancerous tissue* and *malignancy*, and additionally rephrased the prompt to *Can you identify cancer in the image [...]?"*. Table 4 reports F1 scores as percentage-point differences relative to each model's baseline. For ANTONI-$\alpha$, all prompt variations led to improved F1 scores, with *malignancy* yielding the largest gain (+10.2 pp) and *cancerous tissue* and the rephrased prompt showing comparable improvements (+8.2 pp and +9.6 pp, respectively). MedGemma showed a more inconsistent pattern: *malignancy* produced no change, *cancerous tissue* yielded a slight gain (+3.3 pp), while the rephrased prompt led to the largest improvement (+9.5 pp). Notably, ANTONI-$\alpha$ outperformed MedGemma across all prompt formulations.

These findings highlight a well-known issue in the field regarding prompt sensitivity. While ANTONI-$\alpha$ appears more robust to lexical substitution than MedGemma, both models exhibit non-trivial performance variation across semantically equivalent prompts. We leave further investigation of prompt robustness for future research.

| Prompt variant | ANTONI-$\alpha$ | | MedGemma | |
| --- | --- | --- | --- | --- |
| | F1 | $\Delta$ | F1 | $\Delta$ |
| *cancer* (baseline) | .732 | — | .667 | — |
| *cancerous tissue* | .814 | +8.2 | .700 | +3.3 |
| *malignancy* | .833 | +10.2 | .667 | 0.0 |
| *can you identify cancer [...]?* | .828 | +9.6 | .762 | +9.5 |

Table 4: F1 scores for binary cancer classification across prompt variations. $\Delta$ denotes percentage-point difference from each model's baseline prompt (*"cancer"*).

