# OpenReview forum: "Democratising Pathology Co-Pilots: An Open Pipeline and Dataset for Whole-Slide Vision-Language Modelling"
_MIDL.io/2026/Conference — MIDL 2026 Poster_

### Official Review · Reviewer_5LR1 · 2026-01-04

**Confidence:** 3
**Preliminary Rating:** 4
**Final Rating:** 4

**Summary:**

This paper presents a complete open-source framework for building vision-language assistants in computational pathology, addressing the lack of publicly available instruction-tuning data. It introduces Polysome, a pipeline that converts pathology reports into synthetic dialogue, and releases HISTAI-Instruct, a massive dataset of over 1.1 million instruction pairs derived from 24,000 whole-slide images. Using this data, the authors train ANTONI-alpha, a model that processes slide-level feature embeddings directly to outperform generalist models like MedGemma on diagnostic and identification tasks.

**Strengths:**

- Releasing the full pipeline, including the Polysome generation tool, the HISTAI-Instruct dataset, and the model weights, constitutes a significant contribution to the open science community in medical imaging.
- Processing whole-slide images via feature aggregation and attentional pooling addresses the critical resolution bottleneck that limits current models relying on downsampled thumbnails.
- Automated quality control using an "LLM-as-a-Judge" demonstrates a rigorous approach to curating synthetic data and minimizing noise in the training set.
- Scaling experiments provide valuable insights by quantifying how increasing the volume of instruction-tuning data directly correlates with improved performance on downstream tasks.
- Writing is clear and the distinction between existing closed-source efforts and this open initiative is well-articulated.

**Weaknesses:**

- Evaluation is restricted to a held-out internal test set from the HISTAI source, which fails to prove generalization capabilities across different scanners or medical centers like TCGA.
- Comparing ANTONI-alpha against MedGemma feels somewhat unbalanced since the baseline was forced to use low-resolution thumbnails while the proposed method had access to high-resolution feature bags.
- Hallucinations regarding non-visual attributes, such as genetic mutations or IHC results, persist because the training pipeline does not perfectly filter out text descriptions unrelated to the H&E input.
- Reliance on synthetic data generated by another LLM creates a circular dependency where the student model inevitably learns the biases and reasoning patterns of the teacher model rather than ground truth clinical reasoning.

**Detailed Comments:**

- Section 2.1 mentions excluding slides that were "too small" based on a probability threshold, but specifying the exact pixel dimensions or tissue area cutoff would improve reproducibility.
- Figure 2 is conceptually clear, yet adding details about the initialization strategy for the 256 learnable query tokens would be beneficial for those attempting to replicate the architecture.
- It would be interesting to see an analysis of how the model handles "I don't know" scenarios, specifically when asked about features not present in the slide, to better quantify the hallucination risk mentioned in the discussion.
- Citation years (e.g., 2026) indicate this is a future-looking draft, but please ensure all references are updated to their final publication status upon camera-ready submission.

**Justification Of Final Rating:**

The rebuttal addressed most of my concerns in external validation (preliminary COBRA F1 0.785), baseline fairness (MedGemma cannot process PRISM embeddings without retraining), and hallucination analysis (identified "positive compliance bias").

**Justification Of The Preliminary Rating:**

- This paper tackles a major barrier in the field by democratizing access to large-scale instruction-tuning data and capable WSI-level models.
- While the lack of external validation is a notable weakness that prevents a higher score, the methodological soundness of processing native WSI features and the immense utility of the released resources justify a weak accept.
- Addressing the generalization question in the rebuttal would significantly strengthen the case for inclusion.

**Questions To Address In The Rebuttal:**

- Can you provide performance metrics on an external public dataset (e.g., TCGA or CAMELYON) to demonstrate that the model generalizes beyond the HISTAI data distribution?
- How does the performance gap change if the baseline MedGemma is provided with a stronger visual representation, such as the same PRISM embeddings, rather than just a thumbnail?
- What specific steps or heuristics could be added to the Polysome pipeline to better disentangle visual H&E features from non-visual clinical context (genetics/IHC) in the source reports?

---

> ### Author Response · Authors · 2026-01-25
>
> Dear reviewer, thanks for taking the time to review our paper, which has provided us valuable feedback for the camera-ready version. See our responses to your questions below, and please see the adjusted manuscript with the highlighted changes in the rebuttal section.
>
> ---
>
> Q: Can you provide performance metrics on an external public dataset (e.g., TCGA or CAMELYON) to demonstrate that the model generalizes beyond the HISTAI data distribution?
>
> A: Given the feedback from the reviewers we noticed that external validation is an important matter to this study. We take this feedback seriously and have been working for the past week to include a public dataset in our analysis. We used the publicly available COBRA dataset [1], which comprises over 7000 histopathology whole-slide-images related to the diagnosis of basal cell carcinoma skin cancer. Initial results on a subset of 50 cases (50\% cancer) show that ANTONI-$\alpha$ performs better than MedGemma-4B on a classification prompt (cancer yes/no). With an F1 of $.785$ (ANTONI-$\alpha$) versus $.70$ (MedGemma). In this analysis we noted a large sensitivity to different prompts and the effects of reasoning before classification. With these results we plan to extend on the current external dataset with more cases and more elaborate analysis of failure modes, which we plan to describe in detail in the camera-ready version of the paper.
>
> [1] Radboud University Medical Center, "COBRA: Classification Of Basal cell carcinoma, Risky skin cancers and Abnormalities," Registry of Open Data on AWS, 2023. [Online]. Available: https://registry.opendata.aws/cobra. Accessed: 19-01-2026.
>
> ---
>
> Q: How does the performance gap change if the baseline MedGemma is provided with a stronger visual representation, such as the same PRISM embeddings, rather than just a thumbnail?
>
> A: This represents the inherent problem we are trying to solve in the paper. Generally available models such as MedGemma are designed for standard image tokens, which can be extracted from low-resolution images, but cannot handle non-pixel inputs such as PRISM embeddings without retraining. One of the main contributions is the implementation of this domain shift through a projection layer in ANTONI-$\alpha$, and then retraining the LLM component in tandem using these slide-level embeddings. In an attempt to make the comparision more fair, we have also tried splitting up a 2k x 2k thumbnail into multiple 'tiles' of smaller size and supplying these to the model in sequence. This would be comparable to providing multiple images. This increased the amount of visual information to the MedGemma model by a factor of 2-8, but performance was actually reduced to supplying a single low-resolution thumbnail. Therefore, it is not obvious how to optimize the visual input for these models for whole-slide tasks, which exactly points to the benefit of our proposed methods. This particular technical limitation of MedGemma has been made more explicit in Section 3 of the modified manuscript.
>
> ---
>
> Q: What specific steps or heuristics could be added to the Polysome pipeline to better disentangle visual H\&E features from non-visual clinical context (genetics/IHC)?
>
> A: This is an important question for future research, and a couple ideas come to mind. Based on previous research [2], a sequence-to-sequence model can be trained to classify and filter out non-H\&E descriptors. We admit that filtering non-H\&E descriptions should be a separate step in the preprocessing pipeline. The prompts used to generate the conversational attributes in this research include some instructions to ignore non-H\&E features, like in the following form: "*CRITICAL: Do not include any information about clinical history, patient age/gender, specimen measurements (mm/cm), anatomical locations, gross descriptions, or other non-microscopic information, even if provided in the context*". However, as the LLM has to adhere to multiple instructions from the prompt at the same time, it is difficult to verify and quantify how effective this approach is.  We have added a short discussion on this topic to the discussion section of the revised manuscript together with a discussion of a qualitative stress test on hallucination (see answer below).
>
> [2] R. T. Lucassen, T. van de Luijtgaarden, S. P. J. Moonemans, W. A. M. Blokx, and M. Veta, "Preprocessing Pathology Reports for Vision-Language Model Development," in Proc. MICCAI Workshop Comput. Pathol., vol. 254, 2024, pp. 61–71. [Online]. Available: https://proceedings.mlr.press/v254/lucassen24a.html

---

> > ### Author Response · Authors · 2026-01-25
> >
> > Q: It would be interesting to see an analysis of how the model handles ``I don't know'' scenarios...
> >
> > A: We performed a qualitative stress test on a subset of cases to characterize the model's abstention capabilities. The results reveal a clear difference between **clinical reasoning** and **visual grounding**:
> > - The model demonstrates strong emergent guardrails for non-visual attributes. It correctly refuses to predict prognosis (5-year survival), treatment responses (chemotherapy), or definitive molecular status (KRAS, MSI) solely from H\&E images, often citing the need for a more complete clinical picture and additional testing.
> > - Conversely, the model struggles to refute the presence of specific absent visual features. When explicitly prompted about morphological traits (e.g., necrosis, mitotic figures), it exhibits a ``positive compliance bias,'' where it hallucinates their presence rather than stating they are absent.
> > - **Contextual Artifacts:** Occasionally, the model falls back on training priors, referencing non-existent ``pathology reports'' to justify its uncertainty (e.g., when counting lymphocytes). This may be due to incorrect phrasing and references in the generated training data.
> >
> > We have included these transcripts in the Appendix of the modified manuscript as well as an analysis of them in the discussion.
> >
> > ---
> >
> > Q: Specifying the exact pixel dimensions or tissue area cutoff [for small slides] would improve reproducibility.
> >
> > A: Thank you for pointing out this important point for reproducibility. During initial experiments, we indeed did not go into much detail regarding the specific limitations of the used segmentation model. After a manual check of the 40 cases that were not segmented, we found that not all the tissue was specifically too small. To clarify: there were seven significantly out-of-focus slides, and four slides from the source dataset that were shifted along the y-axis, to the point that tissue structures could no longer be recognized. This clarifies why the model missed these cases. The remaining 29 cases consisted of ``micro-biopsies" with an average mean area of $0.57 mm^2$ (HISTAI mixed) and $0.31 mm^2$ (HISTAI skin). These represent a known failure mode for automated segmentation, as models struggle with images with a high background-to-tissue ratio (debris-laden), drowning out the actual tissue signal [3].
> >
> > [3] S. Naghshineh Kani et al., "Tissue Region Segmentation in H&E-Stained and IHC-Stained Pathology Slides of Specimens from Different Origins," medRxiv, 2025. doi: 10.1101/2025.01.16.25320663. [Online]. Available: https://www.medrxiv.org/content/early/2025/01/17/2025.01.16.25320663

---

### Official Review · Reviewer_CVGu · 2026-01-09

**Confidence:** 4
**Preliminary Rating:** 4
**Final Rating:** 4

**Summary:**

The paper presents an open-source, slide-level “copilot” vision-language model for histopathology. To build the VLM, the authors also release Polysome, a standardized tool for large-scale instruction data generation, and introduce HISTAI-Instruct, a multi-category conversational instruction-tuning corpus. The model follows a LLaVA-style pipeline that connects a pathology vision backbone (Virchow to PRISM slide representation) to a medical LLM (MedGemma-4B). Importantly, the paper frames “democratizing” not merely as releasing model weights, but as releasing the entire pipeline (data generation, filtering, and training), which is directionally valuable for the community.

**Strengths:**

1. The manuscript is well structured and easy to follow.
2. Transparency and reproducibility. The work is composed of openly described components and provides an end-to-end pipeline, which lowers the barrier for reproduction and extension. In particular, releasing the Polysome codebase and workflows is valuable for the community.
3. Large-scale instruction data with explicit quality control. The dataset construction includes an LLM-as-a-judge filtering stage and a frequency-based diversification strategy to reduce repetitive prompts, which are sensible steps toward mitigating overfitting and hallucinations

**Weaknesses:**

1. Limited methodological novelty beyond scaling and engineering. The VLM design largely follows established LLaVA-style multimodal projection and training practices; the primary contribution appears to be the open pipeline + large-scale instruction data rather than a novel model architecture. Relatedly, since SlideChat and other recent pathology VLM efforts have released or are releasing similar resources, it would help to more clearly articulate what is uniquely enabled by this work

**Detailed Comments:**

1. Clarify how Polysome generates the seven conversation categories. While Table 1 names the categories, readers would benefit from (i) one concrete, end-to-end example for a single case (input metadata → prompt → generated dialogue), and (ii) a brief explanation of how category-specific constraints are enforced. This would strengthen the argument that Polysome is a key novelty and not merely a scaling tool.

2. Please explicitly reference and discuss Figure 3, 4 in the main text (what exact claim it supports, and what failure mode it illustrates)

3. Provide a per-language breakdown and clarify which splits are used for training. Since the corpus is multilingual but later training appears to use English variants for instruction tuning, please report counts per language and the exact usage in each training stage

**Justification Of Final Rating:**

Thank you for the detailed rebuttal and the revised manuscript. My main concerns have been adequately addressed. In particular, the added clarification of the Polysome pipeline (configuration-driven DAG, category-specific prompt nodes, and the workflow reference) substantially improves the reproducibility and strengthens the paper’s core contribution. I also appreciate that you explicitly discussed Figures 3 and 4 in the main text and clarified the per-language breakdown and dataset split strategy.

Regarding my earlier request for a controlled head-to-head comparison with SlideChat, I acknowledge the authors’ point that a direct comparison may be confounded by differences in training objectives and benchmark focus, and that domain shift effects would be expected when evaluating across mismatched task formulations. Given these constraints, I find the authors’ response reasonable, and the planned external validation (e.g., COBRA) is a constructive next step.

Overall, the manuscript is now clearer and more complete, and I consider my previous concerns resolved. I maintain my weak accept recommendation.

**Justification Of The Preliminary Rating:**

This paper is clearly written and well organized, and I value its emphasis on openness: releasing the full pipeline and code makes the contribution transparent and likely reproducible by the community. However, the test set is relatively small and limited in scope, which leaves open questions about generalization to broader populations and clinical settings. For these reasons, I consider the work a weak accept, promising and reproducible, but in need of stronger, more diverse validation.

**Questions To Address In The Rebuttal:**

1. Evaluation could better reflect the benefits of conversational instruction tuning. The benchmark focuses on Q1–Q3 tasks that are largely “answerable” in a classification-like setting (organ ID, neoplasm detection, and multiple-choice differential diagnosis). It would strengthen the paper to include more evaluation on open-ended multi-turn clinical reasoning/reporting tasks.

2. Limited methodological novelty beyond scaling and engineering. The VLM design largely follows established LLaVA-style multimodal projection and two-stage training practices; the primary contribution appears to be the open pipeline and large-scale instruction data rather than a novel model architecture. Notably, the manuscript itself argues that SlideChat, despite being publicly available, relies on a relatively small cohort (e.g., ~4.2k slides), which may limit generalization. In that case, a controlled head-to-head comparison (e.g., on shared benchmarks and/or under matched training settings) would substantially strengthen the motivation by directly demonstrating the benefit of the proposed scaling and pipeline.

---

> ### Author Response · Authors · 2026-01-25
>
> Dear reviewer, thanks for taking the time to review our paper, which has provided us valuable feedback for the camera-ready version. See our responses to your questions below, and please see the adjusted manuscript with the highlighted changes in the rebuttal section.
>
> ---
> Q: Clarify how Polysome generates the seven conversation categories...
>
> A: Polysome operates as a configuration-driven Directed Acyclic Graph (DAG), enabling users to define complex generation pipelines via JSON. In this study, we utilized a 51-node workflow: a data loader first ingests clinical metadata, which streams into seven parallel text prompt nodes. Category-specific constraints are enforced at this stage via distinct prompts templates bound to each node, strictly isolating the generation logic for tasks like Differential Diagnosis from Clean Report. These outputs then cascade into 42 parallel translation nodes (spanning six languages) before a final aggregation node compiles the dataset. We have added a simplified schematic of the process followed by Polysome to the Appendix of the updated manuscript, and the full configuration is available at the Polysome Github repo (histai-instruct-workflows/histai_instruct_generate_workflow.json). We now also refer to this in the main text of the updated manuscript (Section 2.2)
>
> ---
>
> Q: Please explicitly reference and discuss Figure 3, 4 in the main text (what exact claim it supports, and what failure mode it illustrates)
>
> A: Thanks, we have added a short discussion of these figures in the revised paper at Section 3 (figure 3) and Section 4 (figure 4).
>
> ---
>
> Q: Provide a per-language breakdown and clarify which splits are used for training... report counts per language.
>
> A: We split our dataset on case level, and each case has the same number of conversational attributes and translations (7 conversational attributes in 7 languages each, so 49 attributes per case). Therefore, the split in languages mirrors the overall split of the cases across the train, test and validation split. During quality checks using our LLM-as-a-judge approach, a small number of attributes were removed from individual cases, so some cases may miss one attribute (including multilingual) as explained in the main text. We have now made this explicit in Section 2.2 of the modified manuscript.
>
> ---
>
> Q: It would strengthen the paper to include more evaluation on open-ended multi-turn clinical reasoning/reporting tasks.
>
> A: We strongly agree that open-ended reasoning is the ultimate goal for pathology VLMs. However, the field currently lacks a Gold Standard benchmark for slide-level, open-ended, multi-turn dialogue. Quantitative metrics for open-ended text (like BLEU or ROUGE) correlate poorly with clinical accuracy and semantic similarity, and reliable ``LLM-as-a-Judge'' protocols for whole-slide pathology are still in their infancy.
>
> While this limitation prevents us from including such an evaluation in this study, our group is actively developing benchmarks to address this gap (e.g., the DALPHIN project, \textit{Lab. Invest. 2024}). We view the current work as establishing the necessary training and data generation infrastructure, which will enable the community to move beyond the classification/multiple-choice tasks we presented and towards the complex reasoning you correctly identify as important. A discussion of this has now been included in a new paragraph of Section 5, which also includes a part about multilingual benchmarking.
>
> ---
>
> Q: A controlled head-to-head comparison [with SlideChat]... would substantially strengthen the motivation.
>
> A: We argue that a direct comparison given the available materials is difficult due to a misalignment of training and benchmarking objectives. SlideChat focuses more on reporting standards (e.g. AJCC 2002), grading, and staging (TMS); conversely, Antoni-Alpha is optimized for high-level morphological description and differential diagnosis. Consequently, evaluating either model on the other's benchmark would measure the domain shift rather than intrinsic capability. However, as we are working on additional external validation on the COBRA dataset [1], we will consider including SlideChat in the comparison for the camera-ready version.
>
> Furthermore, given the currently limited organ representation in our training set (predominantly Skin/Breast), we believe the most valuable contribution of this work is not the model weights themselves, but the open-source pipeline. This tool is designed precisely to solve the data scarcity issue, enabling the community (and future work) to generate the diverse datasets required for fair, unified benchmarking and model development.
>
> [1] Radboud University Medical Center, "COBRA: Classification Of Basal cell carcinoma, Risky skin cancers and Abnormalities," Registry of Open Data on AWS, 2023. [Online]. Available: https://registry.opendata.aws/cobra. Accessed: 19-01-2026.

---

### Official Review · Reviewer_Ujv4 · 2026-01-09

**Confidence:** 4
**Preliminary Rating:** 4
**Final Rating:** 5

**Summary:**

The authors study WSI-level vision-language models. They introduce a tool (Polysome) for generating instruction-response pairs from metadata and clinical reports, they apply this tool to the HistAI dataset to generate HistAI-Instruct, and finally they train a vision-language model (ANTONI-$\alpha$) using this HistAI-Instruct dataset.

The vision-language model is based on the LLaVA approach and uses Virchow and PRISM as frozen tile-level and slide-level feature extractors, respectively.

They evaluate their model on three tasks on a held-out internal test set of 317 cases. It is compared to MedGemma, and also to two variants trained on smaller subsets of HistAI-Instruct.

**Strengths:**

- The paper is well written and things are clearly explained overall.
- The studied problem of _slide_-level vision-language models for pathology is particularily interesting.
- The overall methodological approach makes sense and the resulting ANTONI-$\alpha$ model seems to have encouraging performance.
- The entire implementation is made publicly available, and I think that both the Polysome data generation tool, the HistAI-Instruct dataset, and the ANTONI-$\alpha$ training pipeline could be useful to the research community.

**Weaknesses:**

- The evaluation is fairly limited, with just three tasks, 317 cases, and a single baseline model. And, no external validation. However, I'm not sure if that much more can be expected given the quite limited space allowed in the MIDL paper format.
- Some details regarding the training data could be made more clear (see "Questions" below).

**Detailed Comments:**

Questions:
- In Section 2.2 you write that _"To increase linguistic diversity, English instructions were generated first and subsequently translated into six additional languages (Dutch, French, German, Italian, Polish, Spanish). This yielded an unprecedented whole-slide instruction tuning dataset with >1.1M conversational instances"_. Does this mean that there are only 1.1/7 = 0.157 million unique instruction-response pairs in the dataset, with 7 versions for 7 different languages?

- In Section 2.3 you write that _"In the first pretraining stage, [...] We trained the model on the multilingual pathology reports (generated using the clean report task)"_. Does this mean that you here have 7 reports for each of the 18k WSIs in the train set, this is what the model is trained on? Is the performance improved by using all 7 languages instead of just the English reports?

- In Section 2.3 you then also write that _"In the second stage, [...] The training data encompassed the English variants of the tasks summarised in Table 1"_. Why do you only use the English data in this stage, does that give better performance compared to using all 7 languages?

- Also, is there any particular reason why you used Dutch, French, German, Italian, Polish and Spanish? Why not, for example, only Dutch, French and German?  Or, why not using even more languages?

- In Section 3 you write that _"the differential diagnosis task required the model to distinguish between clinically similar options; it was presented with a specific differential of three conditions and prompted to select the most probable diagnosis. To ensure consistent evaluation, we introduced an automated scoring pipeline where the free-text reasoning generated by each model was parsed using Gemini 2.5 Flash (chosen for its reliability and speed) to extract the definitive answer"_. Could you perhaps just expand a bit on this? When prompted to select one of three options, what does your model actually output? Why is this Gemini-based parsing step required?

**Justification Of Final Rating:**

The studied problem is interesting, the paper is well written, the proposed approach makes sense overall, and I definitely think the provided open source implementation could be useful to the research community.

The authors also provided a solid rebuttal with further clarifications. I think this paper should be accepted.

**Justification Of The Preliminary Rating:**

The studied problem is interesting, the paper is well written, the proposed approach makes sense overall, and I definitely think the provided open source implementation could be useful to the research community.

**Questions To Address In The Rebuttal:**

I already have a positive view of this paper, but I would appreciate further clarification on the questions above. I would then consider recommending this paper for an oral presentation.

---

> ### Author Response · Authors · 2026-01-25
> **Clarifications and Additional Motivation in Regards to the Multilingual Data and Training and Differential Diagnosis task**
>
> Dear reviewer, thanks for taking the time to review our paper, which has provided us valuable feedback for the camera-ready version. See our responses to your questions below, and please see the adjusted manuscript with the highlighted changes in the rebuttal section.
>
> ---
>
> Q: Does this mean that there are only 1.1/7 = 0.157 million unique instruction-response pairs in the dataset, with 7 versions for 7 different languages?
>
> A: Correct. While there are ~167k (instead of 157k) unique conversational attributes, we treat the multilingual versions as a form of linguistic data augmentation (the main reason for multilingual training is described below). The number of unique instruction-tuning response pairs has been made more explicit in the modified manuscript in Section 2.2.
>
> ---
>
> Q: Does this mean that you here have 7 reports for each of the 18k WSIs... Is the performance improved by using all 7 languages?
>
> A: Yes, your understanding of the data generation pipeline is correct. Regarding performance: while we have not isolated the specific gain of the multilingual component via an ablation study in this initial release, existing literature in (V)LM training supports the use of translated pairs to improve representation learning (see source 1 and 2 below) due to perceptual and cultural diversity across languages.
>
> We agree that a comparison between multilingual vs. english-only trained models would be scientifically valuable. However, as this requires a full training run from scratch, it was not feasible within the short rebuttal window. We have added a small section in the modified paper acknowledging this trade-off and highlighting it as a key direction for future optimization path of VLM development (See Section 5). In addition we will improve the motivation of multi lingual data by referring to the sources below (see Section 2.3 of the modified manuscript).
>
> [1] T. Nguyen et al., "Multilingual Diversity Improves Vision-Language Representations".
>
> [2] K. Buettner, J. T. Emmerson, and A. Kovashka, "A multimodal recaptioning framework to account for perceptual diversity across languages in vision-language modeling".
>
> (See rebuttal summary for full citations due to character limit).
>
> ---
> Q: Why do you only use the English data in the second stage?
>
> A: We focused on English instruction-tuning because our validation benchmark was created in English. Expanding to multilingual instruction tuning would increase the computational cost seven-fold without a corresponding multilingual benchmark to validate the performance on these languages. We do want to emphasize that our current pipeline and dataset makes multilingual instruction tuning straightforward and we would like to invite the community to think about setting up multilingual evaluation strategies.
>
> ---
>
> Q: Is there any particular reason why you used Dutch, French, German, Italian, Polish and Spanish? Why not, for example, only Dutch, French and German? Or, why not using even more languages?
>
> A: The selection was pragmatic and driven by three key factors. First, we prioritized high-resource languages that are well-represented in the pre-training data of our underlying translation model (Gemma 3). Extending to lower-resource languages could increase the risk of translation errors, particularly regarding sensitive medical terminology. Second, these languages cover a majority of the target clinical user base, and critically, we could perform qualitative sanity checks on these translations within our own research group to verify the outputs. Finally, we aimed to maximize linguistic diversity within our computational budget, drawing the line at seven languages to ensure we delivered a high-quality, verified dataset rather than a larger but potentially noisier one. We have added these motivations to the manuscript (See Section 2.2).
>
> ---
>
> Q: When prompted to select one of three options [in differential diagnosis], what does your model actually output? Why is this Gemini-based parsing step required?
>
> A: The model is prompted with a clinical scenario and three options, appended with a formatting instruction: ``Please provide your final answer in double brackets, like [[diagnosis]].'' Consequently, the model outputs a free-text response that includes clinical reasoning followed by a conclusion. The Gemini-based parsing step was necessary because the models did not consistently adhere to the strict `[[...]]` syntax. Regex frequently missed correct answers, or picked the wrong option, due to minor formatting deviations (e.g., missing brackets, extra punctuation, or verbose concluding sentences). We employed Gemini as a robust semantic parser to normalize these outputs into a structured format. To ensure reliability, we qualitatively validated a subset of these parsings to confirm that the evaluator accurately reflected the model's textual intent. Once again, we have included this additional motivation in the adjusted manuscript (See section 3).

---

> > ### Comment · Reviewer_Ujv4 · 2026-01-31
> >
> > Thank you for the response.
> >
> > The other reviewers are also positive overall, and the authors provided a solid rebuttal with further clarifications.
> >
> > I have increased my score to "5: Strong accept".

---

### Author Rebuttal · Authors · 2026-01-25

**Rebuttal:**

We thank the reviewers for their positive assessment and for recognising the value of our open-source contributions: Polysome, HistAI-Instruct, and ANTONI-$\alpha$. In response to the constructive feedback about the validation of our study, we are working on externally validating our model on the COBRA dataset. Initial results show on a small subset that ANTONI-$\alpha$ (F1: 0.785) outperformed MedGemma-4B (F1: 0.70) in cancer classification tasks (yes/no). However, we aim to increase the size of this external validation set, and also analyse the failure modes of the models in regards to prompt sensitivity and the influence of reasoning preceding label generation. As this validation takes additional time after the rebuttal period, we aim to include it in the camera-ready version.

We also included a qualitative stress test in the updated manuscript to characterise the model's abstention capabilities, identifying a "positive compliance bias" in morphological reporting that provides a clear roadmap for future optimisation. To improve transparency, we have added a simplified end-to-end schematic of the Polysome pipeline to the Appendix and clarified our multilingual data augmentation strategy, justifying the selection of high-resource languages to improve visual-semantic alignment. We have updated the manuscript to include explicit references to all figures, a clarification of data splits across languages, and a more robust discussion on the current lack of "gold standard" benchmarks for open-ended clinical and multilingual reasoning.

These revisions can be found in the revised manuscript attached in the 'supporting materials' of this rebuttal.

Reference list:

T. Nguyen et al., "Multilingual Diversity Improves Vision-Language Representations," in Proc. 38th Annu. Conf. Neural Inf. Process. Syst. (NeurIPS), 2024. [Online]. Available: https://openreview.net/forum?id=1WtEqReCyS

K. Buettner, J. T. Emmerson, and A. Kovashka, "A Multimodal Recaptioning Framework to Account for Perceptual Diversity Across Languages in Vision-Language Modeling," 2025, arXiv:2504.14359. [Online]. Available: https://arxiv.org/abs/2504.14359

**Supporting Material:**

/attachment/0dc86a7f4da250b2eb9fe07266d21774b6cd29c5.pdf

---

### Meta-Review · Area_Chair_Td56 · 2026-02-03

**Recommendation:** Accept (Oral)
**Confidence:** 5

**Metareview:**

The reviewers provided strong positive ratings, minor concerns. The author's rebuttal demonstratesthe strong merit of the paper's impact, and strengthens the paper's contribution. As all evidence stands, the paper can be accepted.

---

### Decision · Program_Chairs · 2026-02-14

Accept (Poster)